# Genome-Wide Search for Gene Mutations Likely Conferring Insecticide Resistance in the Common Bed Bug, *Cimex lectularius*

**DOI:** 10.3390/insects15100737

**Published:** 2024-09-24

**Authors:** Kouhei Toga, Fumiko Kimoto, Hiroki Fujii, Hidemasa Bono

**Affiliations:** 1Laboratory of BioDX, PtBio Co-Creation Research Center, Genome Editing Innovation Center, Hiroshima University, 3-10-23 Kagamiyama, Higashi-Hiroshima City, Hiroshima 739-0046, Japan; togchemi@hiroshima-u.ac.jp; 2Laboratory of Genome Informatics, Graduate School of Integrated Sciences for Life, Hiroshima University, 3-10-23 Kagamiyama, Higashi-Hiroshima City, Hiroshima 739-0046, Japan; 3Research & Development Division, Fumakilla Limited, 1-11-13 Umehara, Hatsukaichi City, Hiroshima 739-0494, Japan

**Keywords:** insecticide resistance, *Cimex lectularius*, genome sequencing, genome editing

## Abstract

**Simple Summary:**

Bed bugs have expanded globally over the past two decades, ausing several health risks. Mutations in their genes allow bed bugs to develop insecticide resistance. However, the extent to which gene mutations exist in the bug genome remains largely unknown because the genomes of resistant strains have not been determined. We accurately sequenced the genomes of both susceptible and resistant strains and compared the gene sequences between the two strains. Several genes with resistance-specific mutations have been identified. These mutations can alter gene function and lead to insecticide resistance.

**Abstract:**

Insecticide resistance in the bed bug *Cimex lectularius* is poorly understood due to the lack of genome sequences for resistant strains. In Japan, we identified a resistant strain of *C. lectularius* that exhibits a higher pyrethroid resistance ratio compared to many previously discovered strains. We sequenced the genomes of the pyrethroid-resistant and susceptible strains using long-read sequencing, resulting in the construction of highly contiguous genomes (N50 of the resistant strain: 2.1 Mb and N50 of the susceptible strain: 1.5 Mb). Gene prediction was performed by BRAKER3, and the functional annotation was performed by the Fanflow4insects workflow. Next, we compared their amino acid sequences to identify gene mutations, identifying 729 mutated transcripts that were specific to the resistant strain. Among them, those defined previously as resistance genes were included. Additionally, enrichment analysis implicated DNA damage response, cell cycle regulation, insulin metabolism, and lysosomes in the development of pyrethroid resistance. Genome editing of these genes can provide insights into the evolution and mechanisms of insecticide resistance. This study expanded the target genes to monitor allele distribution and frequency changes, which will likely contribute to the assessment of resistance levels. These findings highlight the potential of genome-wide approaches to understand insecticide resistance in bed bugs.

## 1. Introduction

Over the past 20 years, bed bugs (*Cimex lectularius* and *Cimex hemipterus*) have spread rapidly globally [1,2]. Bed bugs are obligate hematophagous insects [3,4]. Bed bug bites can cause pruritic rashes [5], folliculitis, and cellulitis due to secondary infections [6]. Although approximately 40 pathogens have been detected in bed bugs from natural populations, no evidence exists that they have been transmitted to humans [7]. Histamine, an aggregated pheromone released by bed bugs, is deposited in house dust and may pose health risks [8,9] Histamine poisoning can occur when ingestion of histamine included in food, leading to rash, urticaria, and edema [10]. Dermatitis occurs by histamine exposure to skin, primally in atopic patients [11].

Insecticide resistance is believed to be a factor contributing to the recent resurgence of bed bugs [10,11]. Infestation was rare in the 1960s because of the use of dichlorodiphenyltrichloroethane (DDT) or other synthetic organic insecticides, such as pyrethroids [2,12]. More recently, spraying with mixtures of pyrethroids and neonicotinoids has been shown to be effective for controlling bed bugs [2,13]. The physiological mechanisms conferring insecticide resistance can be divided into several types: metabolic, penetration, and knockdown resistance [14]. The major groups of enzymes associated with metabolic resistance include cytochrome P450s, glutathione S-transferase (GST), and ATP-binding cassette (ABC) transporters [15]. Cuticle thickening has been implicated in penetration resistance [16], and RNA-seq analysis confirmed that several genes encoding P450s, cuticle proteins (CPRs), and ABC transporters were upregulated in pyrethroid-resistant strains [17,18,19]. Target site insensitivity causes knockdown resistance. To date, amino acid mutations in voltage-gated sodium channels have been observed in the common bed bug *C. lectularius* with pyrethroid resistance [18,20]. Amino acid substitutions from “L to I” or “V to L” in voltage-gated sodium channels have been observed in *C. lectularius* as well as in other insects [21,22]. Recently, resistant strains of *C. lectularius* for organophosphates and carbamates have been found in Japan, and their acetylcholinesterase (AChE, target genes of these two insecticides) possess mutations [23]. However, the genomic information involved in the development of insecticide resistance relies heavily on information from other insects.

Genome sequencing of the common bed bug, *C. lectularius*, has revealed gene sets related to its ecology [24,25]. Previous studies have used short-read sequencing to identify SNPs associated with pyrethroid resistance [26,27]. These studies have identified multiple genomic regions that are likely involved in pyrethroid resistance. However, no study has accurately determined the whole-genome sequence of the resistant strain, and the full extent of mutations in resistant strains remains unknown. In Japan, inquiries into bed bugs have increased since 2000 [28]. In this study, we constructed highly contiguous genomes of susceptible and resistant strains isolated from Japan using long-read sequencing. Next, gene prediction and functional annotation of the assembled genome were performed. We identified amino acid mutations that are potentially related to pyrethroid resistance. This study expanded the target genes to monitor allele distribution and frequency changes, which will likely contribute to the assessment of resistance levels. These findings could highlight the potential of genome-wide approaches to understand insecticide resistance in bed bugs.

## 2. Materials and Methods

### 2.1. Insects

Susceptible strains (Teikyo-u strain) of *C. lectularius* were collected from fields at the Department of Medical Zoology, Research Institute of Endemics, Nagasaki University (Isahaya City, Nagasaki, Japan) by Prof. N. Omori on 22 May 1956 [29]. This strain was maintained at Nagasaki University and Teikyo University and bred and maintained at the Japan Environmental Sanitation Center (Kawasaki City, Kanagawa, Japan) from approximately 1972. This strain was considered insecticide susceptible and was kindly provided to Fumakilla Limited (Chiyoda-ku, Tokyo, Japan) in 2011. The resistant strain (Hiroshima strain) was collected from a hotel in Hiroshima City (Hiroshima, Japan) in July 2010 and maintained by Fumakilla Limited. Bed bugs were reared by feeding mouse blood. All samples used in this study were collected in Japan. Therefore, the international ABS (access and benefit sharing) framework under the Convention on Biological Diversity and the Nagoya Protocol is not applicable.

### 2.2. Insecticide Efficacy Tests

Technical-grade insecticides were used to test the susceptibility. Permethrin (Lot. No. 191001, 97.1%, Sumitomo Chemical Co., Ltd., Tokyo, Japan) is a pyrethroid insecticide, and dinotefuran (Lot. No. K16530422, 99.1%; Mitsui Chemicals Crop & Life Solutions, Inc., Tokyo, Japan) is a neonicotinoid insecticide.

Simplified bioassays were conducted to evaluate the susceptibility of adult *C. lectularius* to the representative insecticides. Bioassays were performed on adults 7–10 days after blood feeding. Adult *C. lectularius* (male:female = 1:1) were held on a piece of tape applied to white paper. Following the topical application method described in WHO [30], an acetone solution (0.4125 µL) of the test insecticide was topically applied to the dorsal mesothorax of each test *C. lectularius* using a hand microapplicator (Burkard Manufacturing Co., Ltd., Rickmansworth, UK). The control group was administered only acetone. Two insecticides, Permethrin and Dinotefuran, were tested using six doses in an insecticide efficacy test, and each replicate contained at least 20 insects. For Permethrin, the Teikyo-u strain was exposed to doses of 0.000391, 0.000782, 0.00156, 0.00313, 0.00625, and 0.0125 μg/individual. Higher doses of 15.6, 31.3, 62.5, 125, 250, and 500 μg/individual were applied to the Hiroshima strain. For Dinotefuran, identical doses of 0.00754, 0.0151, 0.0301, 0.0603, 0.121, and 0.241 μg/individual were used for both susceptible and resistant strains. The treated insects were kept in a plastic cup, the bottom of which was lined with filter paper. Mortality was observed 24 h after treatment, and the half-lethal dose (LD_50_) was calculated using the Bliss’ probit method [31]. A moribund insect that was unable to walk was considered dead. The resistance ratio (RR) was calculated by dividing the LD_50_ of the resistant strain by that of the susceptible strain. Fieller’s equation was used for calculating the 95% confidence intervals [32].

### 2.3. DNA Extraction and Genome Sequencing

The genomic DNA of *C. lectularius* was extracted from the whole body of an adult male using a MagAttract HMW DNA Kit (Qiagen Co., Ltd., Valencia, CA, USA). *C. lectularius* genomic DNA was fragmented to a target size of 10–15 kb using the g-TUBE (Covaris, Woburn, MA, USA). Fragmented DNA was then purified using SMRTbell cleanup beads (PacBio, Menlo Park, CA, USA). A sequence library was prepared using the SMRTbell Prep Kit 3.0 (PacBio) and loaded on the Sequel II sequencing plate 2.0 with on-plate loading concentrations of 90 pM. The libraries were sequenced using Sequel IIe (PacBio). Samples were sequenced using the SMRTCell 8 M tray on the Sequel IIe platform with 30 h of movie time.

### 2.4. De Novo Assembly and Quality Assessment

To remove the mouse genome contamination from HiFi reads of *C. lectularius*, HiFi reads were mapped to the mouse genome sequences (GRCm39) of GENCODE (accessed on 10 January 2024) using Minimap2 [33], and the unmapped reads were used in the subsequent assembly processes. HiFi reads generated from PacBio sequencers were assembled using Hifiasm v0.19.7 using the ‘-primary’ option [34]. Quast v 5.2.0 was used to assess the assembly statistics [35]. Gene set completeness was assessed using the Benchmark of Universal Single-Copy Orthologs (BUSCOs) v5.2.2, with ‘-l insecta_odb10 (10 September 2020)’ or with ‘-l hemiptera_odb10′ (8 January 2024). Merqury v1.3 and Meryl db (k = 21) were used for k-mer-based validation of the genome assembly [36].

Minimap2 was used to align the genome sequences between strains with the ‘-cx asm5′ option. Dot plots were generated using the dotPlotly program with ‘-s -t -m 500 -q 500000 -l’ (https://github.com/tpoorten/dotPlotly, accessed on 8 March 2024) to visualize genomic differences.

### 2.5. RNA Sequencing

Public RNA sequencing data were searched from NCBI using the keywords *Cimex lectularius*. Metadata was downloaded using NCBI SRA Run Selector (https://www.ncbi.nlm.nih.gov/Traces/study/, accessed on 26 February 2024) (Appendix A).

Total RNA was extracted from one adult that was used as a control in 2.2 Insecticide Efficacy Tests using the RNeasy Mini Kit (Qiagen, Hilden, Germany). The extraction was performed in triplicate for both the susceptible and resistant strains. Libraries were prepared using TruSeq stranded mRNA (Illumina, San Diego, CA, USA) and sequenced using NovaSeqX (Illumina), generating 150 bp pair-end reads. Accession numbers are listed in Appendix A.

SRA Toolkit (v3.0.6) (https://github.com/ncbi/sra-tools, accessed on 30 June 2023) was used for SRA retrieval and conversion to FASTQ files. Trim Galore! (v0.6.10) [37] was run for trimming and quality control with option ‘-paired’ options. Low-quality base calls were trimmed using Cutadapt (v3.4) [38] and included Trim Galore! RNA-Seq reads were mapped to the genome assembly using HISAT2 (v2.2.1) with the ‘-q -dta’ option [39]. The SAM files were converted into BAM files using SAM tools [40]. The obtained BAM files were used for gene prediction.

### 2.6. Gene Prediction and Functional Annotation

Repeat regions were identified and soft-masked using RepeatModeler v 2.0.3 and RepeatMasker v4.1.6 in Dfam TEtools v1.88 (https://github.com/Dfam-consortium/TETools) (accessed on 14 March 2024). Gene prediction was performed by BRAKER3 [41] using the soft-masked genome. Besides BAM files, in which RNA-Seq reads were mapped to genome assembly, OrthoDB protein sets of Arthropoda [42] (https://bioinf.uni-greifswald.de/bioinf/partitioned_odb11/, accessed on 17 June 2023) were used as extrinsic evidence when performing BRAKER3. The transcript and amino acid sequences were extracted from the GTF files using GFFread v0.12.7 [43]. Gene set completeness was assessed using BUSCO with ‘-l insecta_odb’ (10 September 2020) and ‘-l hemiptera_odb10′ (8 January 2024). The functional annotation workflow, Fanflow4Insects [44], was modified and used against the reference protein datasets of *Homo sapiens*, *Mus musculus*, *Caenorhabditis elegans*, *Drosophila melanogaster*, hemipteran insects (*Cimex lectularius* (current reference genome Clec2.1), *Acyrthosiphon pisum*, *Daktulosphaira vitifoliae*, *Homalodisca vitripennis*, *Nilaparvata lugens*, *Rhodnius prolixus*), and UniProtKB/Swiss-Prot. Reference protein datasets were retrieved from Ensembl or Ensembl Metazoa (accessed on 5 May 2024). Sequence similarities were searched using GGSEARCH v36.3.8g in the FASTA package (https://fasta.bioch.virginia.edu/; accessed 19 July 2023). A domain search for the Pfam database v35.0 was performed using HMMSCAN in HMMER v3.3.2 (http://hmmer.org/) (accessed on 15 May 2024).

Gene models of susceptible strains were transferred into the genome assembly of resistant strains using Liftoff [45]. The transcripts with different CDS between susceptible and resistant strains were identified by searching for ‘matches_ref_protein: False’ in the GFF files generated by Liftoff. Amino acid sequences of corresponding transcript IDs between susceptible and resistant strains were aligned using MAFFT v 7.525 with ‘-auto’ option [46]. The protein sequence of Clec2.1 was also included in the MAFFT alignment. The corresponding amino acid sequences of Clec2.1 were identified from the results of Fanflow4Insects, as mentioned above. Alignment files were imported using the AlignIO module in Biopython 1.83, and only transcripts with resistant-specific mutations were selected. On the list, if transcripts were missing start and/or stop codons after liftoff, the correctness of the missing codons was manually verified by identifying contig fragmentation or liftoff mistakes. Differences in amino acid sites between the susceptible and resistant strains were visualized on a tree [47]. To characterize the list of transcripts with resistance-specific mutations, enrichment analysis was performed using Metascape (https://metascape.org/, accessed on 24 June 2024), with default settings [48].

## 3. Results

### 3.1. Lethal Effects of Pyrethroid for the Susceptible and Resistant Strains

We evaluated the resistance status of the Hiroshima strain. (Table 1). LD_50_ of susceptible strain (Teikyo-u strain) showed 0.002 µg of permethrin per individual, whereas LD_50_ of Hiroshima strain showed 40.1 µg of permethrin per individual. The resistance ratio (RR) was 19,859, which was higher than that of many previously discovered strains [49]. Hereafter, we refer to the Teikyo-u strain as the susceptible strain and the Hiroshima strain as the resistant strain.

### 3.2. De Novo Genome Assembly

We generated primary contigs with genome sizes of 644,683,587 bp and 614,893,704 bp for the susceptible and resistant strains, respectively (Table 2). Indicators for genome contiguity were of higher quality than those of the current reference genome Clec2.1 (Table 2). The contig N50 lengths for the susceptible and resistant strains were 1,530,193 bp and 2,132,290 bp. These assembly metrics were of higher quality than those of the current reference genome Clec2.1 (Table 2). The number of contigs in both strains was lower than that in the Clec2.1 reference genome. The BUSCO completeness in both strains was comparable to that of the Clec2.1 reference genome. K-mer-based evaluation demonstrated high base accuracy and completeness of primary contigs in susceptible and resistant strains. The susceptible strain showed a quality value (QV) of 57.0 and completeness of 97.8%, while the resistant strain had a QV of 56.9 and completeness of 94.9%. The K-mer spectrum analysis revealed two peaks in both strains, indicating the construction of a diploid genome assembly (Figure 1a,b). No counts were observed that could be considered sequence errors. We examined the genomic differences between susceptible and resistant strains using dot-plot visualization (Figure 1c). Linear plots confirmed the high similarity between the susceptible and resistant strains.

### 3.3. Gene Prediction and Functional Annotation

Because repetitive sequences may affect gene prediction accuracy [50], these sequences in the genome assembly were searched using RepeatModeler and RepeatMasker. In both strains, approximately 50% of the genomic regions contained repeat sequences (Appendix A, shown in bases masked). These values are slightly higher than those of the current reference genome, Clec 2.1. BRAKER3 predicted 13,116 and 13,104 protein-coding genes in the susceptible and resistant strains, respectively (Table 3). Gene annotations and protein sequences are available at figsShare (https://doi.org/10.6084/m9.figshare.26362135.v2, accessed on 29 August 2024). The BUSCO completeness values for both strains were comparable to Clec2.1. Gene annotations of the susceptible strains were transferred to the resistant strain using Liftoff [45], identifying the corresponding genes between both strains (12,724 protein-coding genes and 20,151 transcripts) (Table 3). The BUSCO completeness of the resistant strains after liftoff did not decline substantially, indicating that the transfer of gene predictions was successful (Table 3). Functional annotation was added to the protein-coding genes of the susceptible strains using Fanflow4Insects, which included GGSEARCH and HMMSCAN. The number of gene hits for each reference protein is shown in Table 4. Of 13,116 genes in susceptible strains, 99% protein-coding genes (13,010/13,116) were successfully annotated by GGSEARCH and HMMSCAN. The genes that could not be annotated were classified as hypothetical proteins (Table 4). The complete results of Fanflow4Insects are presented in Appendix A. We performed a BUSCO analysis using databases for glires and primates. If significant contamination were present in the assembly constructed in this study, the BUSCO completeness scores would be higher. However, the BUSCO analysis of assemblies from susceptible strains showed completeness values of 4.9% for glires and 3.5% for primates. These results suggest that the assemblies of susceptible strains are not composed of genomes derived from either mice or humans.

### 3.4. The Search for Mutated Genes Related to Insecticide Resistance

As a previous report suggested, point mutations are related to the acquisition of pyrethroid resistance in bed bugs [21]. These genes can be identified on a genome-wide scale by comparing the protein sequences of susceptible and resistant strains. Liftoff revealed 3938 transcripts with amino acid mismatches between susceptible and resistant strains. These amino acid sequences were aligned with those of Clec2.1 using MAFFT, and the alignments among the three strains were compared. Clec2.1 genome comes from a susceptible strain [24]. The MAFFT alignment revealed 729 transcripts with resistance-specific mutations (Appendix A). We investigated the number of mutation sites in the amino acid sequence of each transcript (Figure 2a, Appendix A). The majority of the transcripts (496 of 729) had one mutation in their amino acid sequences. *Laminin subunit beta 1* had the largest number of mutated sites (54). To verify the reliability of the gene list, we searched for known or candidate genes with resistance-specific mutations, including sodium voltage-gated channels, AChE, gamma-aminobutyric acid (GABA) receptors, nicotinic acetylcholine receptors (nAChR), P450s, ABC transporters, GSTs, and CPRs [14]. Transcripts with resistance-specific mutations were detected in all these genes, except for the GABA receptor (Figure 2b–g). No studies have identified mutations in *C. lectularius* nAChR, a target gene of neonicotinoid insecticides. Although we determined the resistance ratio of the neonicotinoid insecticide (dinotefuran) between the susceptible and resistant strains (Appendix A), the Hiroshima strain did not show neonicotinoid resistance (resistance ratio: 1.11).

Enrichment analysis was performed on 729 transcripts to characterize the genes (Figure 3 and Appendix A). Enrichment analysis using Metascape was conducted using *Homo sapiens* gene ID or *D. melanogaster* gene ID, which were described in Appendix A. The molecular functions of a large number of genes are evolutionarily conserved among species, and there is abundant and detailed information on human gene function. For this reason, we used human gene IDs for enrichment analysis to infer molecular functions in insects. Using *Homo sapiens* gene ID, we identified several GO terms potentially related to the development of insecticide resistance, as described below. DNA metabolic process, mitotic cell cycle process, insulin metabolic process, and Lysosome were enriched (−log10 (*p*-value) > 6) (Figure 3a). Gene lists associated with the GO terms are shown in Appendix A. Forty-two genes were included in the “DNA metabolic process” category. Metascape clustered several similar ontologies, and the summarized results are shown in the GroupID column in Appendix A. The “DNA metabolic process” category included multiple terms related to DNA damage and repair, such as “DNA replication”, “DNA repair”, “DNA damage response”, “double-strand break repair”, and “recombinational repair” (Appendix A). Fifty-two genes were clustered under the GO term “mitotic cell cycle process” (Appendix A). Among these genes, CCNB2 (cyclin B) and ORC1 are activated during M and G1 phases of the cell cycle, respectively [51,52]. Nine genes clustered under the GO term “insulin metabolic process” (Appendix A). Among these genes, CPE (Carboxypeptidase E) is involved in the proteolytic processing of proinsulin peptides in humans [53]. Thirteen genes were clustered under the GO term “Lysosome” (Appendix A). Intracellular proteases respond to insecticide stress in resistant strains to maintain cellular health [54]. As expected, the cathepsin genes (CTSB and CTSD) were included among the 13 genes. Additionally, components of lysosomes, such as SLC11A2, were included. SLC11A2 is a divalent metal transporter that is a component of lysosomes [55].

When using *D. melanogaster* gene ID, “Lysosome” was enriched, similar to the results obtained using human gene IDs (Figure 3b, Appendix A). GO terms included cathepsin. Tetraspanins (Tsp29Fa and Tsp42Ee) were also identified. The enriched GO term “glucose transmembrane transport” was related to the “insulin metabolic process” enriched when using human gene ID, as these were associated with nutrient response [56].

## 4. Discussion

We successfully constructed a highly contiguous genome assembly of susceptible and resistant strains of *C*. *lectularius* and identified mutations that may be related to the development of insecticide resistance. The present study confirmed the mutations conferring pyrethroid resistance in the amino acids of the sodium voltage-gated channel, as expected from previous reports in *C*. *lectularius*. We confirmed mutations in other target genes that confer neonicotinoid or OPs resistance (AChE and nAChR). Although mutations in AChE have been found in *Cimex* species [23], no reports have described mutations in the nAChR. However, we were unable to confirm neonicotinoid resistance in this study. Mutations found in nAChR may not be associated with neonicotinoid reception. Genes related to metabolic resistance (CYP450 and ABC transporters) or penetration resistance (CPRs) were highly expressed in resistant strains. To the best of our knowledge, this is the first study to report mutations in these genes.

A total of 729 transcripts with resistance-specific mutations were identified. We found “DNA metabolic process”, “Lysosome”, “Insulin metabolic process” and “glucose transmembrane transport” and “Mitotic cell cycle process” as enriched GO terms in the 729 transcripts. These biological processes may be related to the physiological adaptation of the resistant strains, as discussed below.

### 4.1. DNA Metabolic Process

Enrichment analysis revealed that resistance-specific mutations were present in many genes associated with DNA damage, repair, and replication. Bifenthrin, a pyrethroid insecticide, causes DNA damage in Sf9 cells [57]. In *Anopheles coluzzii*, the expression of DNA repair-related genes increased after exposure to pyrethroids [58]. Mutations identified in this study may promote DNA damage repair in resistant strains, leading to adaptations to insecticide exposure.

### 4.2. Lysosome

In the house fly *Musca domestica* and red flour beetle *Tribolium castaneum*, lysosomal protease activity generally increases in response to exposure to pyrethroids and DDT [54]. Actually, protease functioning in the lysosome, cathepsin B, was present in the result of enrichment analysis. Mutations were also observed in the lysosomal components of tetraspanin (*Tsp29Fa* and *Tsp42Ee*). In *D. melanogaster*, *Tsp42Ee* is primarily located in the retina [59], and its mutation leads to light-induced retinal degeneration. Whether the tetraspanin mutation found in this study is related to insecticide resistance through the visual system may provide a target for understanding the ecology and evolution of resistant strains.

### 4.3. Insulin Metabolic Process and Glucose Transmembrane Transport

Enrichment analysis Insulin signaling is conserved between mammals and insects, and regulates circulating sugar levels [56]. Therefore, we discuss insulin metabolic processes and glucose transmembrane transport in the same section.

Mutations were present in glucose and trehalose transporters (*Tret1-1*, *pippin*, and *nebu*) (Appendix A). *Tret1-1* expression is upregulated in response to starvation in *D*. *melanogaster* and *T*. *castaneum* [60,61,62], leading to the efficient uptake of trehalose and glucose. *Pippin* regulates the selective transport of carbohydrates into the brain [63], but its response to starvation remains unknown. Mutations in these transporters may contribute to resistance to starvation.

When the human gene ID was used for enrichment analysis, the insulin metabolic process was enriched. Components of insulin signaling are regulated in response to starvation in *D. melanogaster* [56]. Therefore, “insulin metabolic process” and “glucose transmembrane transport” may be associated with each other. Additionally, in *D. melanogaster*, 114 mutations in insulin-signaling genes have been found in DDT-resistant strains [64]. In cotton leafworm *Spodoptera litura*, exposure to azadirachtin, a tetranortriterpenoid found in neem trees (*Azadirachta indica*), deactivates insulin signaling [65]. Oral treatment of pyridalyl upregulated the expression of insulin receptor in the olive fruit fly *Bacterocera oleae* [66]. These results suggest the connection between insulin signaling and insecticide exposure. In conclusion, modifying insulin signaling may be a common feature in conferring insecticide resistance in insects.

### 4.4. Mitotic Cell Cycle Process

DNA damage is monitored during cell cycle progression, and the cell cycle is arrested until the DNA damage is repaired [67]. Mutations were observed in 52 genes associated with the mitotic cell cycle. Among these, ORC1 has been extensively studied with respect to cell cycle regulation [68]. ORC1 plays a central role in the assembly of the pre-replication complex to initiate DNA replication, and its activation is regulated during the cell cycle [69]. Mutations observed in insecticide-resistant strains may allow strict regulation of the cell cycle, preventing daughter cells from inheriting damaged DNA in response to insecticide exposure.

### 4.5. The Relationships among GO Terms Enriched in This Study

The GO terms identified in this study (DNA metabolic process, insulin metabolic process, glucose transmembrane transport, lysosomes, and mitotic cell cycle process) may be related to cell cycle regulation. DNA metabolic processes play crucial roles in cell-cycle progression and arrest of cell cycle [67]. Cell cycle progression depends on sugars such as glucose [70]. Insulin signaling regulates the cell cycle (i.e., cell growth and cell cycle progression) by activating mTOR signaling [71]. Furthermore, lysosomes function as nutrient sensors in response to amino acid inputs [72], suggesting an association between lysosomes and the response to nutrients in *C. lectularus*. Investigating the phenotypic functions of the mutated genes will be a challenge in the future. Genome editing is an effective method for elucidating molecular mechanisms underlying insecticide resistance.

### 4.6. Limitations of This Study

The mutations identified in this study may be rare, depending on the populations surveyed. In bed bugs, population genetics to detect alleles with mutation have been carried out on the voltage-gated sodium channel α-subunit gene [20]. The functional contributions of the mutations to pyrethroid resistance levels could be better elucidated by comparing genes from a larger number of common bed bug strains. Furthermore, strict evaluation for pyrethroid resistance levels may need a population genetic approach to clarify the allele frequencies.

At this stage, we can only link the mutated genes identified in this study to functions known in other insects, as it has not yet been demonstrated how these mutations contribute to changes in insect physiology. While these mutations are likely to enhance efficiency in relevant physiological processes, gene functional analyses are necessary to fully understand how these mutations contribute to the development of insecticide resistance.

## 5. Conclusions

Through genome-wide search for gene mutations in resistant strains of *C. lectularius*, we identified 729 mutant transcripts that have scarcely been studied in common bed bugs by comparing the sequences of protein-coding genes. These genes may enable the development of insecticide resistance in the resistant strains. A population genetics approach may clarify the extent of functional contributions of these mutations. In bed bugs, population genetics to detect alleles with mutation have been carried out on the voltage-gated sodium channel α-subunit gene [20]. This study contributes to the expansion of target genes to detect the distribution and changes in allele frequencies within populations. Distribution and changes in allele frequency may also be helpful for assessing the degree of insecticide resistance. In addition, genome editing of these genes may be an effective method for understanding the evolution and physiological mechanisms involved in the acquisition of insecticide resistance.

## Figures and Tables

**Figure 1 insects-15-00737-f001:**
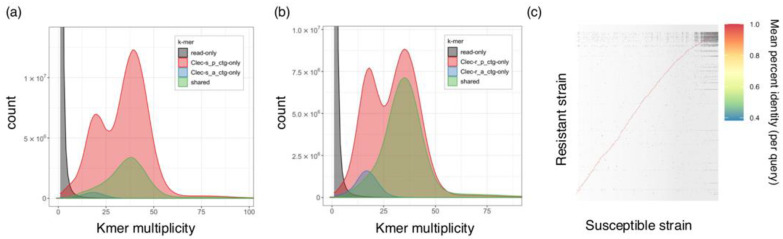
K-mer assembly evaluation by Merqury and dot plot between susceptible and resistant strains. (**a**) K-mer spectrum assembly plot generated by Merqury of the susceptible strain and (**b**) resistant strain. Primary or alternate contig are indicated by different colors. (**c**) Dot-plot between susceptible strain and resistant strain. Color legend indicates the mean percent identity (per query).

**Figure 2 insects-15-00737-f002:**
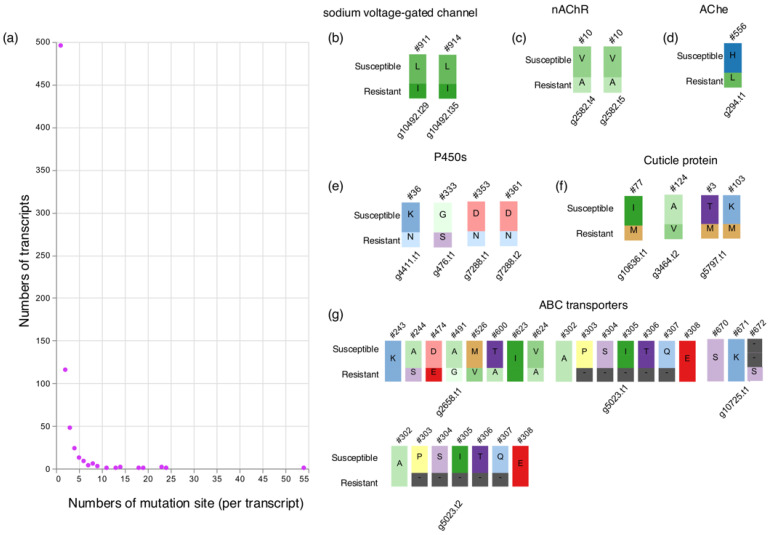
Transcripts with mutations in the resistant strain. (**a**) The number of mutation sites per transcript is shown by circles. (**b**–**g**) Mutation sites of candidate resistance genes are shown with different amino acids. “Susceptible” refers to susceptible strain sequenced in this study and Clec2.1. Gene IDs are indicated by ‘g’ followed by a number, and transcript variations are denoted by ‘t’ followed by a number. In (**b**), two mutated sites corresponded to the sites of 925 in housefly *Musca domestica*.

**Figure 3 insects-15-00737-f003:**
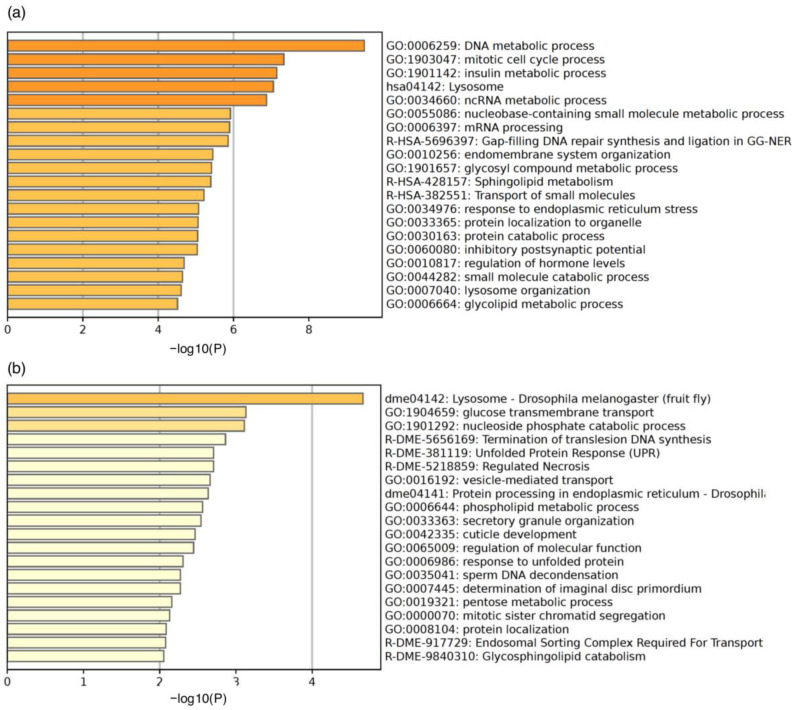
Results of enrichment analysis by Metascape. (**a**) Results using human gene ID and (**b**) results using fly gene ID are shown, respectively. Bar graphs are colored differently according to the value of −log10(P).

**Table 1 insects-15-00737-t001:** The LD_50_ value and resistance ratio of each strain to two insecticides.

Insecticide	Strain	Number of Specimens	Slope ± SE	LD_50_ (µg/Individual)	LD_50_ 95% CI	df	*p*-Value	RR	RR 95% CI
Permethrin	Teikyo-u	120	3.1 ± 1.08	0.0020	0.0015–0.0028	4	0.77	-	-
	Hiroshima	120	1.3 ± 0.39	40.1	24.4–66.0	4	0.73	19,859	10,559–38,993

SE: standard error; LD_50_: the dose of insecticide that is lethal to 50% of the test specimens; df: degree of freedom; RR: LD_50_ (Hiroshima strain)/LD_50_ (Teikyo-u strain); CI: confidence interval.

**Table 2 insects-15-00737-t002:** Comparison of assembly statistics with the current reference genome.

	Parameter	Susceptible	Resistant	Clec2.1
Assembly statistics	Length (bp)	644,683,587	614,893,704	510,848,842
	contig N50 (bp)	1,530,193	2,132,290	552,400
	contig L50	124	85	285
	No. of contigs	1324	1092	2868
	Largest contig	12,214,012	9,042,003	6,600,950
	GC%	35.08	35.19	34.82
BUSCO results (OrthoDB v10 insect database)	Complete [single, duplicated]	99.3% [98.1%, 1.2%]	99.2% [97.8%, 1.4%]	99.3% [98.1%, 1.2%]
	Fragmented	0.3%	0.2%	0.3%
	Missing	0.4%	0.6%	0.4%
BUSCO results (OrthoDB v10 hemiptera database)	Complete (single, duplicated)	99.4% [97.8%, 1.6%]	99.1% [97.4%, 1.7%]	99.6% [98.2%, 1.4%]
	Fragmented	0.2%	0.2%	0.1%
	Missing	0.4%	0.7%	0.3%

**Table 3 insects-15-00737-t003:** Statistics of protein-coding gene prediction.

BRAKER and Liftoff		Susceptible	Resistant	Clec2.1 *	Resistant (after Liftoff)
	The number of protein-coding gene	13,116	13,104	11,949	12,724
	The number of transcript	20,928	20,890	24,194	20,151
BUSCO results (OrthoDB v10 insect database)	Complete [single, duplicated]	98.4% [69.1%, 29.3%]	98.3% [68.0%, 30.3%]	99.7% [65.9%, 33.8%]	97.3% [68.5%, 28.8%]
	Fragmented	0.1%	0.3%	0.3%	0.4%
	Missing	1.5%	1.4%	0.0%	2.3%
BUSCO results (OrthoDB v10 hemiptera database)	Complete (single, duplicated)	98.1% [65.3%, 32.8%]	97.8% [64.9%, 32.9%]	99.5% [60.3%, 39.2%]	96.9% [65.7%, 31.2%]
	Fragmented	0.0%	0.1%	0.0%	0.1%
	Missing	1.9%	2.1%	0.5%	3.0%

* The data of Clec2.1 were referred to Ensembl statistics and protein fasta file.

**Table 4 insects-15-00737-t004:** Summary of functional annotation results.

Annotation Category	Reference (Protein ID)	Number of Gene Hits
	*Homo_sapiens*	9913
Gene annotation by top hit (GGSEARCH)	*Mus_musculus*	9738
	*Caenorhabditis_elegans*	8323
	*Drosophila_melanogaster*	9585
	*Cimex_lectularius* (Clec2.1)	11,933
	*Acyrthosiphon_pisum*	10,269
	*Daktulosphaira_vitifoliae*	10,151
	*Homalodisca_vitripennis*	10,481
	*Nilaparvata_lugens*	10,578
	*Rhodnius_prolixus*	10,098
	UniProtKB/Swiss-Prot	10,828
	at least one of the above	12,993
Genes annotated only by HMMSCAN	Pfam domain database	17
All genes annotated at protein level		13,010
Hypothetical protein	No hit	106
Total		13,116

## Data Availability

Assembled sequences and raw sequence data were deposited in DDBJ under the umbrella BioProjects PRJDB18520 and PRJDB18521 for susceptible and resistant strains, respectively. Primary and alternate contigs of the susceptible strain were deposited under the accession numbers BAAFRD010000001-BAAFRD010001324 and BAAFRE010000001-BAAFRE010006538, respectively. Primary and alternate contigs of the resistant strain were deposited in the DDBJ under the accession numbers BAAFRF010000001-BAAFRF010001092 and BAAFRG010000001-BAAFRG010004661, respectively. Raw sequence reads were deposited in the DDBJ Sequence Read Archive under the accession numbers DRA018917 (HiFi reads of susceptible strains), DRA018918 (HiFi reads of resistant strains), and DRA018916 (RNA-Seq reads). The results of gene prediction and functional annotation are available at figshare (https://doi.org/10.6084/m9.figshare.26362135.v2, accessed on 29 August 2024).

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
