# Peer review of "Genome-Wide Search for Gene Mutations Likely Conferring Insecticide Resistance in the Common Bed Bug, Cimex lectularius"

_insects, 2024, doi:10.3390/insects15100737_

Round 1

Reviewer 1 Report

Comments and Suggestions for Authors

In this study entitled “Genome-wide Search for Gene Mutations likely Conferring Insecticide Resistance in the Common Bed Bug, Cimex lectularius”, authors proposed genome-wide identification of gene mutations in the genomes of susceptible and pyrethroid-resistant Cimex lectularius using long-read sequencing and compared their amino acid sequences to identify gene mutations. Gene mutation-based research of common Bed Bug is essential due to the demands of health risk, and insecticide resistance development. It could pave the way for mitigating resistance-related insect management. I suggest that the editor consider publishing this manuscript after the following revisions.   

The manuscript needs revisions. The following suggestions are provided only to help the authors improve on the currently submitted version of the manuscript. 

Comment 1: 

Abstract: In the abstract section, please provide the findings of this study rather than the research gap. The research gap is supposed to be in the introduction section. Please rewrite the abstract.

Comment 2: 

Please make all the tables appropriate. Below the table, the elaboration is mentioned, i.e., CI, SE, LD. The text (Lines 188-192) mentions that the genome size is bp, whereas the Table mentions Mb. I am confused. Please make sure the unit of the genome and other parameters is unique.  

Comment 3:

In Figure 1 a,b, c, X-axis, and Y-axis font sizes should same and legible.

Comment 4: 

The conservation of insulin signaling between mammals and insects focuses on regulating circulating sugar levels. It mentions the presence of mutations in glucose and trehalose transporters, which are believed to contribute to resistance to starvation. The section also highlights the enrichment of the insulin metabolic process when human gene IDs are used for analysis, suggesting that insulin signaling modifications may play a role in insecticide resistance.  The point is that Pippins are used as regulators of carbohydrate transport into the brain. Could you clarify what 'pippins' refer to, as this term might not be widely recognized in this context?" 

Comment 5: 

Please provide additional evidence or references supporting the connection between insulin signaling modifications and insecticide resistance in insects, especially in other species beyond Drosophila melanogaster.

Comment 6: How was the enrichment analysis conducted, particularly the methodology behind using human gene IDs?

Comment 7: In the reference section, please maintain consistency. Reference should be uniform according to the prescribed journal. Here, a few journal names are abbreviated, and some are elaborated. Please update the appropriate reference format of the insects journal. 

In light of the aforementioned points, I recommend that the manuscript be revised. Addressing these points will greatly improve the manuscript and could have a valuable impact on the field of insecticide resistance research. 

Reviewer 2 Report

Comments and Suggestions for Authors

Dear Toga et al.,

I have identified the following deficiencies for improvement in the manuscript:

1.       Doses Tested: The authors tested six different doses for each strain (line 100).   It is essential to specify which six doses were used in the study.

2.       Degree of Freedom (df) Values: For the toxicology bioassay experiments, the degree of freedom (df) values should be included.   I recommend adding these values to Table 1.

3.       P-Value Calculation: In the regression equation fitting for virulence, the P-value tests the chi-square statistic.   A P-value greater than 0.05 indicates that the difference is not significant, which is preferable.   Conversely, a P-value less than 0.05 suggests a poor fit of the equation, indicating a substantial deviation between observed and theoretical values.   Please include the P-values in Table 1.

4.       Discussion Section: The discussion does not fully address the results.   Please expand on how the mutations in resistance-specific genes identified in your study contribute to bed bug resistance development, incorporating insights from previous research

Reviewer 3 Report

Comments and Suggestions for Authors

The manuscript reports on genomic and transcriptomic differences between a resistant and susceptible strain of bed bug. The authors do a good just obtaining very nice genomes of both susceptible and resistant strains, which is nice for the community resources. I like that both genomic and expression data was used to find genes that may be involved in resistance. However, I was confused on how the RNA reads from NCBI were incorporated and what the control adult was used for. Please see my more specific comments below.

line 40: what are the health risks of histamine in house dust?

line 43: what is used to control bed bugs in modern times?

Methods:

-2.1. The resistant strains were collected in 2010 and maintained in the lab thereafter. Was this strain challenged with insecticide since 2010? Do you think it has lost some of the genetic changes that cause or contribute to resistance?

- 2.2 what is the final concentration of permethrin and neonicotinoid that the insect comes in contact with?  line 93 states that you evaluated susceptibility but did you also use increasing concentrations to evaluate the level of resistance in the resistant strain?

-2.3. Can more details be added to the sequencing methods and library prep? What was the movie time, library concentrations and sizes, etc.?

- 2.4. Mouse contamination because they were fed on mice? Please clarify.

 -2.5. Can you explain what RNA reads were used for the experiment from NCBI? Were these reads from the same strains as your study. Was the RNA collected after exposure to the same insecticides? Line 30: I don't understand why one adult was used as a control. Was it resistant or susceptible and to which insecticide?

Results:

Table 1 implies that concentration bioassays were conducted, but I don't see those details in the methods.

Table 4. Are the human and mouse hits from contamination?

Figure 2. Really interesting results and highlighted many types of resistance genes that contribute to resistance in bed bugs. 

Figure 3. Can you explain why you used human gene ID for an insect?

As stated in 4.6., it would be really interesting to see this type of work at the population and geographical scale!

Comments on the Quality of English Language

Authors did a good job on the English component. Only minor editorial problems were detected.

Round 2

Reviewer 1 Report

Comments and Suggestions for Authors

The original manuscript has been polished, and now this edited manuscript seems to have improved to a level suitable for publication.

Reviewer 3 Report

Comments and Suggestions for Authors

The authors have addressed my previous concerns.